# Manipulation of fractionalized charge in the metastable topologically entangled state of a doped Wigner crystal

Anze Mraz [1,2] ✉, Michele Diego [1], Andrej Kranjec[1], Jaka Vodeb [1], Peter Karpov [3], Yaroslav Gerasimenko [1,4], Jan Ravnik[1], Yevhenii Vaskivskyi [1,5], Rok Venturini [1,5], Viktor Kabanov [1], Benjamin Lipovšek[2], Marko Topič [2], Igor Vaskivskyi [1,4] & Dragan Mihailovic [1,4,5]

Metastability of many-body quantum states is rare and still poorly understood. An exceptional example is the low-temperature metallic state of the layered dichalcogenide 1T-TaS$_2$ in which electronic order is frozen after external excitation. Here we visualize the microscopic dynamics of injected charges in the metastable state using a multiple-tip scanning tunnelling microscope. We observe non-thermal formation of a metastable network of dislocations interconnected by domain walls, that leads to macroscopic robustness of the state to external thermal perturbations, such as small applied currents. With higher currents, we observe annihilation of dislocations following topological rules, accompanied with a change of macroscopic electrical resistance. Modelling carrier injection into a Wigner crystal reveals the origin of formation of fractionalized, topologically entangled networks, which defines the spatial fabric through which single particle excitations propagate. The possibility of manipulating topological entanglement of such networks suggests the way forward in the search for elusive metastable states in quantum many body systems.

While ground states in superconductors and Bose–Einstein condensates are robust to single particle excitations, finding the true metastability of macroscopic quantum states has proved to be a challenge. In contrast, topological states are globally protected, and are robust to local fluctuations, while potentially supporting metastability. Such states have been suggested to be candidates for quantum computation and memory devices, and are thus also of practical interest. Well-known quantum examples are hybrid semiconductor–superconductor nanowires and topological insulators in contact with a superconductor[1,2], or edge dislocation cores in a topological superconductor[3], but systems derived from such ideas have so far proved challenging to fabricate and control in real materials[4]. On the other hand, topological objects such as magnetic skyrmions[5–7], or domain walls (DWs) in ferroelectric[8] or magnetic nanowires[9] can more easily be manipulated allowing data to be controlled classically by the application of a current, and are already used in memory devices.

Another more recent system relies on the formation of a 'hidden' (H) metastable state, with an intricate chiral domain wall network (Fig. 1d)[10], which can be created, manipulated and destroyed by light[11] or external electric current[12–14]. Practical memory devices based on this phenomenon have already demonstrated ultralow energy dissipation,

[1]Department of Complex Matter, Jozef Stefan Institute, Jamova 39, SI-1000 Ljubljana, Slovenia. [2]Faculty for Electrical Engineering, University of Ljubljana, Tržaška 25, SI-1000 Ljubljana, Slovenia. [3]Arnold Sommerfeld Center for Theoretical Physics, Ludwig Maximilian University, München, Germany. [4]CENN Nanocenter, Jamova 39, SI-1000 Ljubljana, Slovenia. [5]Faculty for Mathematics and Physics, University of Ljubljana, Jadranska 19, SI-1000 Ljubljana, Slovenia. ✉e-mail: anze.mraz@ijs.si

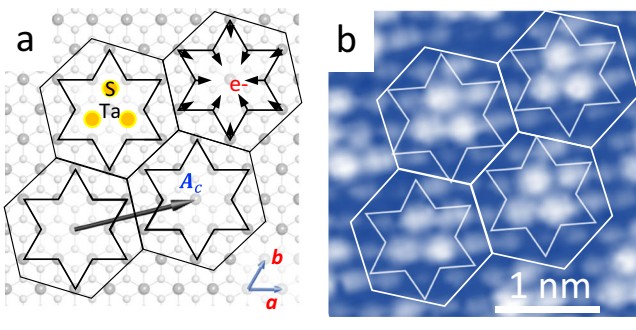

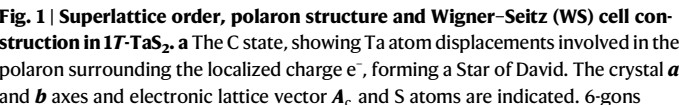

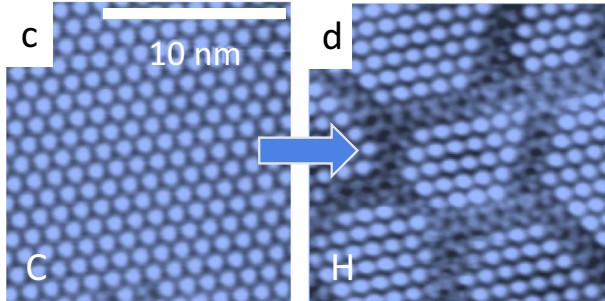

**Fig. 1 | Superlattice order, polaron structure and Wigner−Seitz (WS) cell construction in 1T-TaS₂. a** The C state, showing Ta atom displacements involved in the polaron surrounding the localized charge e⁻, forming a Star of David. The crystal *a* and *b* axes and electronic lattice vector $A_c$ and S atoms are indicated. 6-gons represent WS cell construction for the C state. **b** A scanning tunnelling microscope (STM) image corresponding to **a** with visible *S* orbitals (bias voltage −0.2 V, tunnelling current 100 pA). **c, d** STM images of the C and H states, respectively.

high speed, and high-contrast resistance switching[15–18]. Yet the origin of the unique metastable H state in 1T-TaS₂[10,11,19,20], the origin of its topological structure[10,21], and the charge transport mechanism that exhibits a large observed resistance drop in the metastable state are still open questions[19,22–25]. The material itself (1T-TaS₂) is of fundamental interest as a charge density wave system in the Mott−Wigner limit[26–28]. It supports multiple high-temperature phases, including an incommensurate (IC) phase, a nearly commensurate (NC) domain phase[29,30], and a commensurate (C) insulating phase[31] with frustrated spin order[32] at low temperatures. (full equilibrium and non-equilibrium phase diagrams are described in the Supplementary Note 1). Because of the electron−phonon interaction in 1T-TaS₂, the kinetic energy of electrons is strongly suppressed, resulting in a propensity for their localization[28]. The localized electrons cause polaronic ionic lattice displacements approximately in the form of a Star of David (SoD) in the C state (Fig. 1a, b) with a superlattice unit cell size $\sqrt{13}a \times \sqrt{13}a$, where *a* is the lattice constant, with one electron per primitive Wigner−Seitz (WS) cell (Fig. 1a–c).

Here we explore the dynamics and microscopic structure of metastable electronic networks in the C superlattice of 1T-TaS₂ created by surface current injection into the C ground state[13,33]. For this purpose, we use multiple independently positioned tips in a low-temperature scanning tunnelling microscope (STM), whose positioning on the surface of the crystal is aided by an integrated scanning electron microscope (SEM). This enables us to mimic a practical device, and for the first time microscopically ascertain the absence of conventional CDW sliding behaviour in response to an electric field at low temperatures. The domain networks created in the current path between two closely spaced STM tip electrodes on the crystal surface are analysed in terms of a WS cell construction which enables us to investigate the topological rules and observe charge fractionalization within the networks created by charge injection. While the undistorted WS lattice is hexagonal, we find that the emergent domain structure contains defects mostly in the form of pentagons/heptagon pairs. The STM images reveal the formation of two types of defects at the domain wall crossings: trivial defects, where no dislocation is formed in the WS superlattice and their Burger's vector sum around the vertex is zero; and non-trivial ones, for which a dislocation forms in the WS superlattice with a non-zero Burger's vector[10]. The non-trivial topological defects at the domain wall junctions−which are homotopically equivalent to crystal dislocations−protect the mesoscopic network structure from external perturbations. Correlated dislocations are spatially separated and (classically) topologically entangled within the networks. In our experiments, we can directly monitor the dislocation annihilation dynamics, which is experimentally linked to observed bulk resistivity switching. Device thermal modelling of the 2-tip experiment shows that the switching between states is non-thermal and does not occur via the high-temperature NC state, but directly from the H to the C state. Monte-Carlo (MC) simulations of gradual charge injection based on the Mott−Wigner lattice gas model reveal the topological entanglement rules for the formation of extended networks, providing remarkable insight into the creation of intricate fractionally charged networks as a result of charge injection into the Wigner crystal state. The combination of experiments and modelling reveals very unconventional charge dynamics in the metastable H state of 1T-TaS₂.

## Results

### Network reconfigurations induced by lateral current

Figure 2a shows an in-situ SEM image of the experiment at 4 K with a schematic diagram of the electrical current path between STM tips #1 and #2, positioned ~2.7 μm apart. Unlike in conventional STM imaging[13,33], where the electrical current passes vertically through the tip and the sample to the ground, here the current flows laterally between two STM tips. A third STM tip (#3) is used to obtain local density of states (LDOS) images of the area between the contacts (Fig. 2a) with a non-perturbing current. In the local state approximation, where there is a direct link between the presence of a polaron and the LDOS, for an electron state with energy $E_i$, the LDOS as measured by STM current of tip #3 is given by[34]: $\rho_{\text{local}}(E, \mathbf{r}) \propto \Sigma_{i=1,N} |\psi_i(E_i, \mathbf{r})|^2 \delta(E - E_i)$. Integrating $\rho_{\text{local}}(E, \mathbf{r})$ over $E$ in the interval $\pm \frac{\Delta}{2}$, where $\Delta$ is the full bandwidth of state $E_i$, we obtain a real-space charge density map $\rho_{\text{local}}(\mathbf{r})$. The total effective charge is given by $q = \int \rho(\mathbf{r}) d\mathbf{r}$, where the area $A$ of integration is defined by a superlattice WS cell construction, as shown in Fig. 1b for the C state. The STM can thus be used to investigate microscopically how $\rho(\mathbf{r})$ patterns evolve under the influence of current (see Supplementary Note 5 for a more detailed explanation).

The voltage−current (*V*−*I*) curve shown in Fig. 2b shows the resistance switching at 4 K in response to the injected current pulses through tips #1 and #2, and describes the experimental protocol of the STM measurements by tip #3 presented in Fig. 2c. We start the experiment in the high resistance C state, and ramp up the 'write' (W) current $I_{12}$ (Fig. 2b, red triangles) using 50 μs pulses. The *V*−*I* curve deviates from ohmic behaviour and starts to saturate above ~0.4 mA. Eventually, above a threshold of $I_W^T \sim 4.8$ mA an abrupt drop of resistance occurs, which signifies that the sample has switched to the low resistance H state. The corresponding STM images for the initial high resistance C state and the switched low resistance H state obtained with tip #3 are shown in Fig. 2c panels ⓪ and ①, respectively. In the initial state ⓪, we observe uniform C order, and after a 'write' sequence, a characteristic domain pattern of the H state in panel ① is observed. In the second step, we perform an 'erase' (E) sequence to investigate how the low resistance H state responds to current. We ramp the E current $I_{12}$ (Fig. 2b, blue squares) while simultaneously monitoring the step-by-step reordering of the charge configuration with tip #3. A selection of STM images from the E sequence is shown in Fig. 2c panels ② − ⑥, while the whole E sequence is shown in

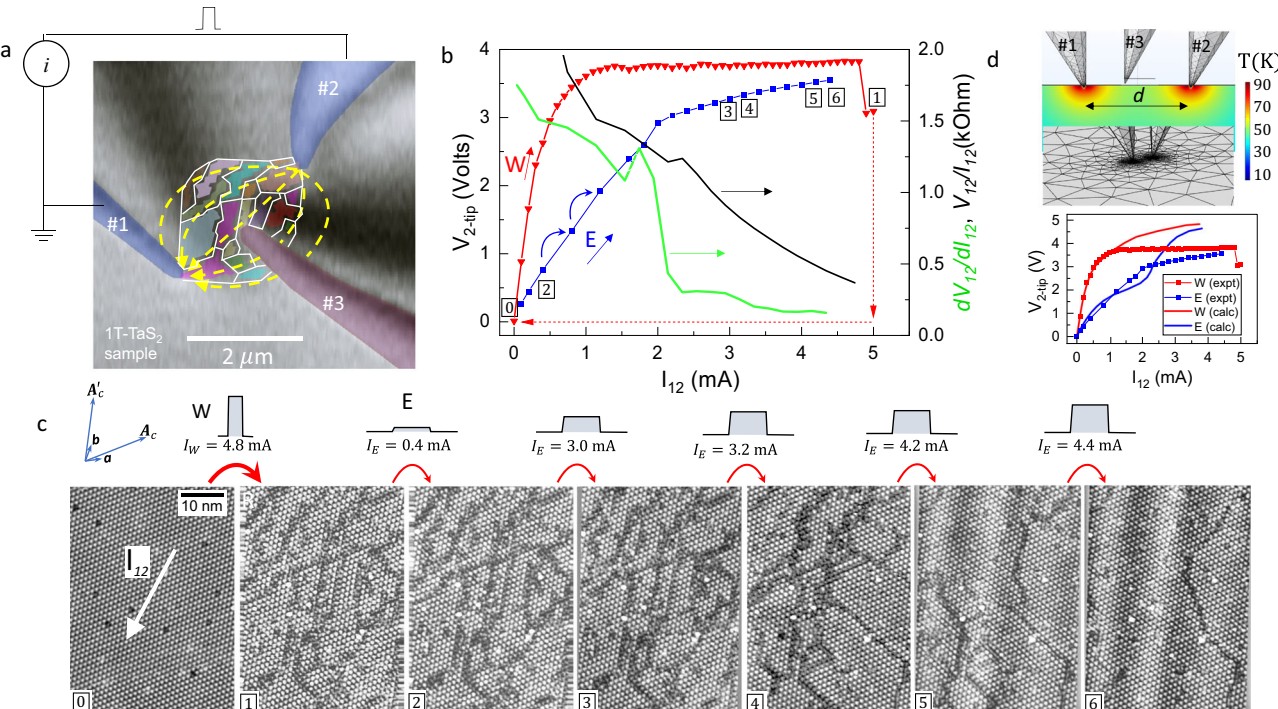

**Fig. 2 | Creation and domains reconfiguration by electrical current. a** In-situ scanning electron microscope image of the experimental setup with the three tips used in the measurements. The domain structure is schematically indicated with coloured domains (not to scale). The current path is shown schematically with yellow arrows. Tips #1 and #2 are for sourcing of the current and #3 is for scanning tunnelling microscope (STM) imaging. **b** V–I curve during the switching cycle: First, switching to the H state is performed by ramping up the W current from zero up to $I_{12}$ = 5 mA between tips #1 and #2 (red triangles). Switching to the H state is observed at ~4.8 mA. Erasing back to the C state is performed by ramping up the E current from zero (blue squares). The slope $\frac{dV_{12}}{dI_{12}}$ (green), and resistance $\frac{V_{12}}{I_{12}}$ (black) curves are also shown. **c** STM images measured by tip #3, the direction of current $I_{12}$ is marked in panel ⓪; panels ⓪ and ① show the C and H states respectively, panels ②–⑥ correspond to the E sequence shown in **b** at a certain current value marked above the panels. Note the white streaks running diagonally in panels ⑤ and ⑥, attributed to lattice strain. **d** The thermal map calculated using finite element calculation (top panel, colour scale is on the right) and the calculated V–I curves obtained by the model without any adjustable parameters (bottom panel) in comparison to the experimental V–I curve from **b** for both the W (red) and E (blue) procedure.

Supplementary Fig. 2 and Supplementary Movie 1, as well the E sequence of a repeated experiment on a different sample can be seen in Supplementary Figs. 6–10 and by Supplementary Movie 2. The slope $\frac{dV_{12}}{dI_{12}}$, and the resistance $R = \frac{V_{12}}{I_{12}}$ are also plotted in Fig. 2b. We see that as $I_E$ increases, the domain configuration is stable at low current, and resistance is nearly ohmic up to a threshold current $I_E^T \simeq 2$ mA (Fig. 2b, blue squares). Above $I_E^T$ we observe a rapid change of slope, and rapid domain reconfiguration can be seen above 3.2 mA, shown in Fig. 2c panels ④–⑥, accompanied by another smaller change of slope at 3.2 mA seen from $\frac{dV_{12}}{dI_{12}}$ in Fig. 2b. For the E sequence an area of the sample with identifiable imperfections was specifically chosen (i) to register the image position, and (ii) to determine if there is any correlation between the created domain structure and the imperfections. Note the conspicuous absence of conventional CDW sliding motion[35,36] at any $I_E$. We note that the measurements reveal a preferential direction for the formation of domain walls along the direction of current, and an accompanying strain, which is visible as light streaks along the direction of the current in the topographic images (see Fig. 2c ⑤–⑥), with more examples in Supplementary Figs. 2 and 4.

An important aspect of the experiment is to show the effect of heating, which was thus far considered to be responsible for the creation of mosaics in the C-CDW structure. A finite element model calculation of Joule heating in response to the current $I_{12}$ at 4 K shown in Fig. 2d (see Supplementary Note 9 for calculation details) reveals that during the W process, in which the current is highest, the sample temperature below the tips #1 and #2 never exceeds 90 K. At tip position #3, where the STM images are taken, the excess temperature

does not exceed 50 K. Thus, the sample temperature is always well below the nearest phase transition temperature (220 K), from which we conclude that the observed network growth is not a result of a quench through a high-temperature phase transition, but is caused directly by current injection via a fundamentally different mechanism[13,37,38].

## Structure analysis of the metastable state

To analyse the topological structure of the observed networks, we first accurately determine the polaron positions both within the domains and the DWs, relative to the underlying crystal lattice for the STM images in Fig. 2c, and perform a WS tesselation (Fig. 3a) for some of the images (see Supplementary Note 8 for details). The effective charge is expressed as a filling fraction for site $i$, defined as $f_i = f_C(A_C/A_i)$, where $A_i$ is the area of the $i$-th cell, $A_C$ is the area of the C superlattice cell calibrated from the experimental data, and $f_C$ is the filling fraction of C state. The resulting structure is very revealing: DWs are identifiable as 6-gons (red or blue) whose charge density differs from the reference density of the C superlattice (white, or near-white). 5-gon/7-gon *pairs*, which define a dislocation, appear at some (but not all) DW junctions, as explicitly shown in Fig. 3b (5-gons are brown, 7-gons are green). The Burger's vector $\vec{B}$, by definition, runs along the line joining the 5-gon and the 7-gon, as shown in the zoom-ins in Fig. 3c. We find that pairs of dislocations with opposite $\vec{B}$ often appear along the direction of the applied switching current (indicated in Fig. 2c), but not always. No isolated disclinations, which correspond to *single* 5-gons or 7-gons, are ever observed.

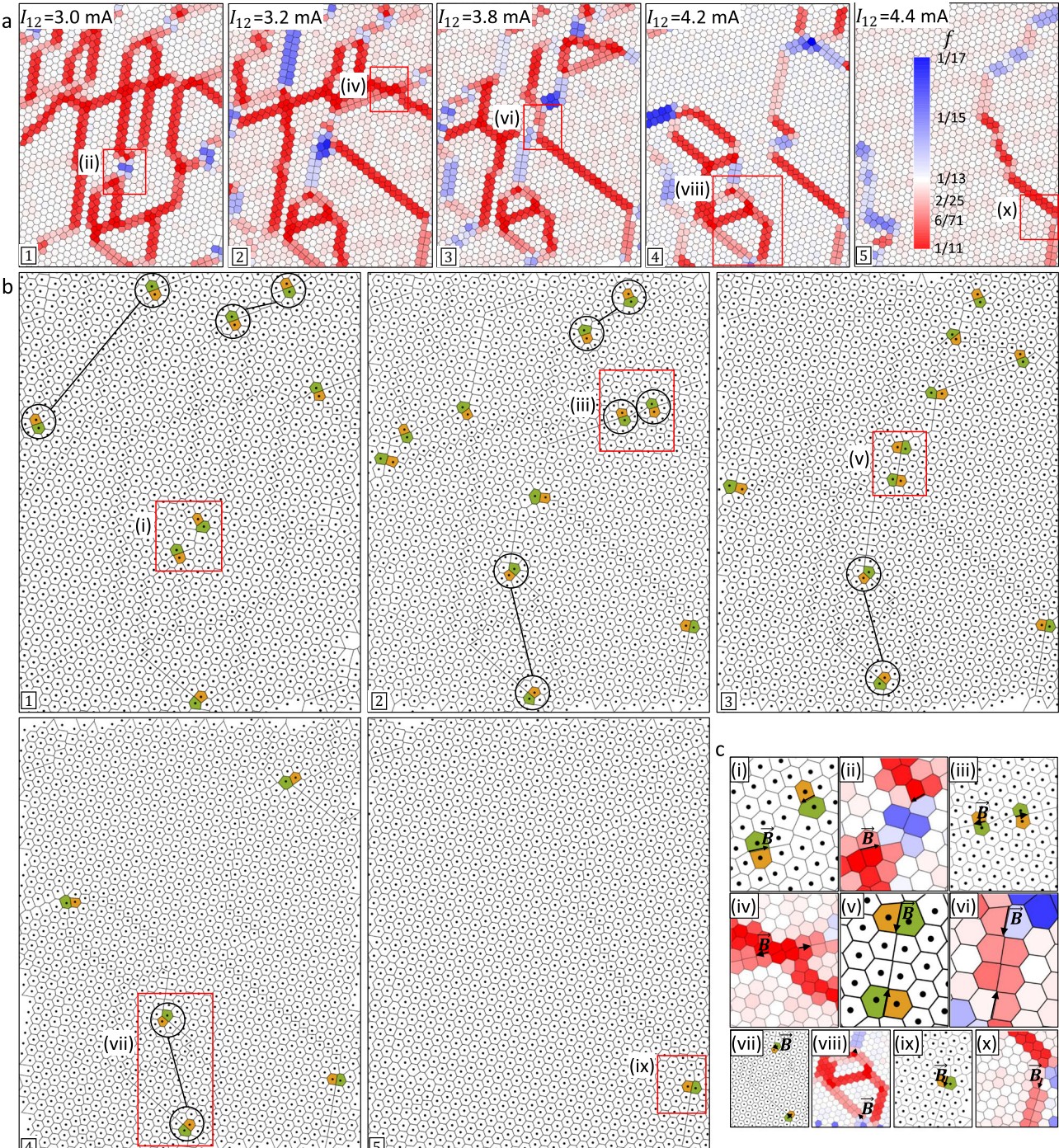

**Fig. 3 | Topology of the network structure in response to an injected current.**
**a** Wigner–Seitz (WS) construction shows the density map or filling fraction $f_i$ of each individual cell for some of the images of the E sequence shown in Fig. 2c, with the value of the current pulse marked. The legend for $f$ is shown in the inset in panel ⑤. **b** A WS map of dislocations, explicitly showing pentagon-heptagon pairs (orange and green, respectively), with panel numbers corresponding to (**a**). Pairs of correlated dislocations are indicated by the joined black circles. The red rectangles numbered (i)–(x) in both **a** and **b** define the areas that are analysed in (**c**). **c** Zoom-in of the red rectangle areas showing in more detail the dislocation pairs (i, iii, v, vii and ix) and corresponding density maps (ii, iv, vi, viii and x). The corresponding Burgers vectors **B** are shown by black arrows.

The dislocation dynamics in the E sequence shown in Fig. 3b are particularly striking. Initially, there is no motion up to 3 mA. At $I_E = 3.2$ mA, shown by the transition from panel ① to ②, we first begin to detect motion of some dislocations, but their total number is conserved. Thereafter, in panels ③–⑤, the dislocations start to annihilate *pairwise*, and eventually, only one dislocation remains in panel ⑤. We note that the observed dislocation dynamics correlate well with the switching of resistance at 3.2 mA ($\frac{dV_{12}}{dI_{12}}$ in Fig. 2b), while switching at 2 mA is not directly recorded with the STM. This is not unexpected, since we are observing only a very small part of the macroscopic domain wall network (see Supplementary Fig. 3). Each time the experimental cycle is repeated, the observed pattern is different (see

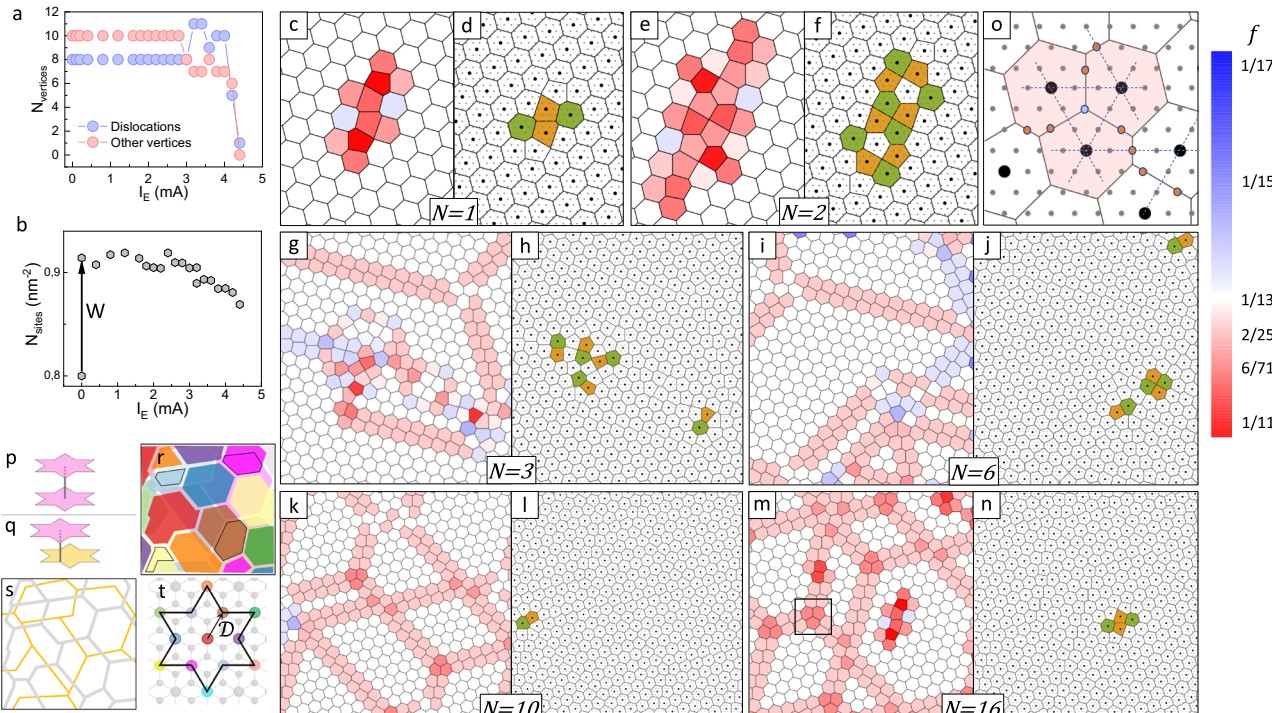

**Fig. 4 | Defect dynamics and modelling of charges injected into a Wigner crystal (WC) superlattice. a** The number of dislocations (blue), and the number of topologically trivial vertices (red) as a function of $I_E$. The data is obtained by counting the number of polarons in the image. The counting error is smaller than the statistical fluctuations of the data. **b)** the total number of Wigner–Seitz (WS) cells per unit area, calibrated to the underlying crystal lattice. The error in the site density is estimated as <1% and the error bars should be smaller than the scatter of the data in the plot. **c** The effect of a single electron ($N = 1$) on the WC lattice predicted by the Monte–Carlo simulation of the charge lattice gas model. The density or filling fraction $f$ scale is shown by the colour bar on the far right. **d** The dislocations due to single electron ($N = 1$) are explicitly shown via pentagon–heptagon pairs (orange and green, respectively). **e, f** Distortion, density and dislocations caused by a pair of electrons ($N = 2$).. **g–n** Model predictions for $N = 3$, 6, 10 and 16 electrons: distortion, density and dislocations. **o** Detailed structure of a vertex, highlighted by the black square in panel **m** showing shared atoms (light red and blue circles), where for each hexagon $f = 6/71$. **p, q** Examples of pairwise and displaced stacking of polarons between adjacent layers, respectively. **r** Schematic representation of stacking of domains between two layers, with the respective displacement colour of each domain shown in (**t**). Occasional pairwise stacking of charge density waves between two layers is highlighted (black line), while other domains have some sort of displaced stacking as in (**q**). **s** Same schematic as in **r**, but displaying only the domain walls for better clarity. Grey and yellow lines represent domain walls in the top and bottom layers, respectively. **t** Colour map of displacement vectors **D**.

Supplementary Fig. 4 and Supplementary Figs. 6–10), from which we deduce that lattice imperfections do not appear to play a significant role in the formation of a specific pattern. A count of the dislocations and trivial vertices is shown as a function of $I_E$ in Fig. 4a. Their number correlates with the drop in resistance at 2–3 mA, as shown in Fig. 2b. Note that the total number does not need to be conserved, as the system is not in equilibrium, and in any case, we are only observing a small part of the network.

In Fig. 4b we see that after the W sequence, the total number of sites $N_{sites}$ in the image increases from 0.8 nm$^{-2}$ (Fig. 2c, panel ⓪) to 0.92 nm$^{-2}$ (Fig. 2c, panel ①), which—in the localized state limit—is consistent with an increase of localized carrier density of ~15%. Thereafter, $N_{sites}$ is approximately constant until dislocations begin to annihilate during the E sequence, dropping to ~0.87 nm$^{-2}$ (Fig. 2c, panel ⑥), consistent with carriers trapped in the two remaining DWs.

### Modelling dislocation networks

The dynamics of such interacting electrons on the verge of localization can be modelled in terms of 'polaronic' Wigner crystal dynamics[21,28,39–42]. The injection of carriers ($e$ or $h$) can be very effectively investigated by introducing additional charges into a commensurate, sparsely filled triangular lattice, using a charge lattice gas (CLG) model Hamiltonian[21,28], $H = \Sigma_{i,j} V_{i,j} n_i n_j$, where the occupation numbers $n_i$ and $n_j$ are 0 or 1, $V_{i,j} = \frac{V_0 \exp(-\frac{r_{ij}}{r_0})}{r_{ij}}$ is the Yukawa screening potential,

$V_0 = \frac{e^2}{\epsilon_0 a}$ (CGS units), and $r_{ij} = |r_i - r_j|$, where $r_i$ is the position of the $i$-th polaron, $r_0$ is the screening radius, $a$ is the lattice constant, and $\epsilon_0$ is the dielectric constant. A commensurate structure occurs when the cosine rule is satisfied for the electron filling $f$, $\frac{1}{f} = x^2 + y^2 + xy$, where $x$ and $y$ are integers that represent primitive vectors along the crystal lattice $\boldsymbol{a}$ and $\boldsymbol{b}$ axes, respectively, where $f$ ranges from ½ to 1/16 in different compounds[28]. The model has been shown to be very effective in describing the phase diagram of a large number of 2D CDW systems at different fillings[28]. For the C superlattice in $1T$-TaS$_2$, $x = 3$ and $y = 1$, giving $f = \frac{1}{13}$ (Fig. 1a, b). When a single additional injected electron ($N = 1$) localizes within the C state, it distorts the superlattice to accommodate an extra WS cell, forming a pair of dislocations with equal and opposite Burgers vectors in the process (Fig. 4d). The range of the distortion extends beyond the two dislocations (Fig. 4c) and is approximately determined by $r_0$. When the injected particle density increases (shown for $N = 2$ to 16 in Fig. 4e–n), the charges become further topologically entangled, the effects of injected charge become non-local, and the DWs are intertwined with dislocations in a non-trivial way. The charge is classically entangled in the sense that a single charge cannot be removed without affecting the network on an extended scale. Conversely, the addition of a single charge results in an extended highly non-trivial modification of the network. This makes the system highly non-linear with respect to doping. The resulting charged network comprises topologically trivial and non-trivial

vertices connected by DWs, where individual dislocations are spatially separated, as shown in Fig. 4h or j, for example. The network growth process stops short of a Kosterlitz–Thouless transition to an isotropic state[43], where dislocations dissociate into individual disclinations, and long-range orientational correlations are lost[42], but rather, remains in a hexatic phase in which the orientational order is retained[44].

The charge density of the distorted cells can be expressed in terms of $f$, as a geometrically defined finite set of fractional numbers. To calculate $f$ in the discrete model, we note that within each such WS cell, there are $N_I$ atomic sites. In addition, $N_E$ atoms may be shared between two cells, and $N_J$ atoms are shared by three cells, when they lie at a junction of 3 DWs as is seen in the example in Fig. 4o, which is a zoom-in of the highlighted square in Fig. 4m (We ignore higher order DW junctions). Here $N_E = \frac{1}{2}n_e$ and $N_J = \frac{1}{3}n_J$, where $n_E$ and $n_J$ are integers that represent the number of occurrences of each type of site. Typically, $n_J \neq 0$ at vertices, and $n_E \neq 0$ when the bisection between two centres of adjacent cells is along a crystal lattice direction (Fig. 4o). The filling fraction for each cell is thus given by $f = 1/(N_I + N_E + N_J)$. The trivial vertex cells (Fig. 4o) commonly have $f = \frac{6}{71}$ ($= 1/11.83$). The DWs themselves have a different specific charge per unit length, most commonly $f = \frac{2}{25}$ ($= 1/12.5$), and a unique well-defined microscopic structure in relation to the underlying atomic crystal lattice[13,45]. Both positive and negative charge densities relative to the C state can occur, and the number of charge density fractions is finite because they are defined by the geometric construction in relation to the atomic lattice (discussed in more detail in Supplementary Note 7). As in the experiment (Fig. 3a, b), the nearby dislocations appear or disappear in pairs. We also observe instances where pairs of **B**s on *neighbouring* dislocations do not sum to 0 (e.g. Figs. 3c(i) and 4h), and compliance with the **B** sum rule may involve multiple dislocations (Fig. 4f). The calculations also reveal pathological examples that globally violate the sum rule, that is, $\Sigma_i \boldsymbol{B}_i \neq 0$ even for an ensemble of patches that are totally surrounded by the C state (see Supplementary Fig. 12), which is consistent with the fact that the system is not in equilibrium. A direct consequence of the non-locality of the injected charge in the fractionalized network is that fractionalized charge can propagate an arbitrary distance away (e.g. Figs. 3c (i), and 4h, j), leading to non-local interactions over large distances.

## Discussion

DWs and dislocations may be discussed in the continuum limit as local spatial modulations of the CDW order parameter[46], but charged defects are not included in the model, so it cannot account for the doping behaviour. The observed phenomenology is quite different from the conventional spontaneous nucleation and growth of domains after a first-order transition, in which mutually spatially uncorrelated domains emerge through a first-order phase transition[47]. We show that the presently observed transition is non-thermal, proceeding as a sequence of local, topologically defined steps, resulting in a strongly correlated, entangled state with non-local interactions. Since the network is charged it experiences a force if an electric field is applied. However, the microscopically observed absence of DW sliding motion in response to a lateral current implies that conventional soliton charge transport mechanisms are not relevant here[48]. The dislocations in the polaronic structure revealed by the WS construction seem to be responsible for the remarkable stability of the domain state at smaller lateral currents, which also has bearing for practical applications, providing non-volatility at low temperatures[15,16]. The network dynamics is slow on the timescale of single particle (SP) motion, but the two cannot be easily separated: The SP quantum interference of correlated electrons within single-layer confined 1T-TaS₂ artificial nanostructures was recently examined in detail by STM, revealing SP trajectories that are strongly modified from free-electron motion and appear in the form of quantum interferences with signatures of quantum scars and chaotic orbits[34]. Such SP quantum interferences may be expected to be present also here, leading to intricate topologically defined quantum probability density networks, that define the SP current paths. Network reconfigurations manifest themselves in step-like resistivity jumps[12], which are related to the Devil's staircase energy level spectrum of the incommensurate domain state[49]. These resistivity jumps and observed quantum interferences[34] reflect the sensitivity of the SP motion to the configuration of the network structure which acts as a manipulable, temporally evolving 'fabric' through which the current-carrying electrons propagate. An investigation of quantum interferences, such as recently observed in real space for topological corner states with fractional charge in graphene[50,51], would contribute to a better understanding of the quantum nature of the fractional states in the H state of 1T-TaS₂.

When discussing macroscopic transport properties, an important aspect is the interaction between neighbouring layers, which strongly depends on the relative stacking of the network structures in different layers. In the insulating C ground state, pairwise stacking of polarons along the c-axis is indicated[52–54], as shown schematically in Fig. 4p. On the other hand, STM experiments in the H domain state have shown that vertices and DWs tend to avoid lying exactly on top of each other in adjacent layers. The vertices typically appear near the middle of a domain in the next layer (shown schematically in Fig. 4r and s)[13], which is consistent with the minimization of the Coulomb repulsion between charged DWs and vertices in adjacent layers. The pairwise stacking is thus perturbed, and the polarons are displaced with respect to each other in adjacent layers, as shown schematically in Fig. 4q. C-like pairwise stacking can occasionally occur if the displacement of domains (as per the colour map in Fig. 4t) in adjacent layers lines up, as shown schematically in Fig. 4r by the highlighted areas, but mostly pairwise stacking is absent. The LDOS, probed by single-tip STS was previously shown to be both gapped and gapless[13,37,45] for different stacking configurations, from which we may surmise that the origin of the metallic behaviour in the H state comes from transport through gapless domain regions that do not have C state stacking. This is consistent with bulk X-ray analysis that suggests rearrangement of the charge and orbital order in the H state in the direction perpendicular to the TaS₂-layers away from pairwise layer stacking[20]. However, we note that it is not necessary to invoke inter-layer transport for the structure to be conducting when domains are present.

The domain structure is also expected to cause buckling of the underlying atomic lattice. This has been shown to cause band states that were previously far from $E_F$ in the undoped C state to be brought to the Fermi level, resulting in metallic conduction and superconductivity[55]. Domain-induced buckling may be expected to have a similar effect[45]. In fact, we can see evidence of a long-range strain developing as a result of lattice buckling at the DWs in the form of long wavelength LDOS modulations running approximately parallel to the DWs in Fig. 2c (panels ⑤, ⑥).

The spatial charge fractionalization and entanglement described by the presented model used to analyse the domain dynamics in the commensurate state of 1T-TaS₂ can be considered to be a universal phenomenon which accompanies domain formation in triangular electronic superlattices. The microscopically observed response to the electric field described here is fundamentally different from CDW sliding, conventionally considered as the favourite mechanism for describing the response of CDWs to an external electric field. The formation of networks also introduces complex mesoscopic lattice deformations which have a profound effect on interlayer stacking, and emergent macroscopic properties, such as current flow. We note that, unlike other fractionally charged quasiparticles, such as anyons and quarks, here the effect of fractionalization can be directly observed in real space (Figs. 3b and 4c−n). The fractional charge may be observable in tunnelling experiments[50,51]. The advance in understanding the origin of the unusual textures that lead to metastability, fractionalization, long-range entanglement and unusual charge transport may lead to

the design of new metastable materials, opening opportunities for developing fast, energy-efficient memory technology based on topological defect manipulation in quasi-2D Van der Waals materials[15].

## Methods

The 1$T$-TaS$_2$ single crystals were synthesized using the chemical vapour transport method. The samples for all STM measurements were cleaved in an ultra-high vacuum. STM was performed in an Omicron LT Nanoprobe at 4 K. STM imaging was performed in constant current mode with the STM tip biased at $-0.8$ V with respect to the sample and the tunnelling current set to 1 nA unless stated otherwise. The drifting of the STM scanner was corrected by FFT peaks of CDW modulation of 1$T$-TaS$_2$ using Scanning Probe Imaging Processor (SPIP) software. For the STM $V-I$ measurements, the two outer tungsten tips $\sim 2.7\,\mu$m apart were first placed in tunnelling contact and then advanced into the sample until an approximately linear $V-I$ curve was reached, (up to ~40 nm beyond the contact point). For the 4-contact STM measurements, two inner tips are used to measure the voltage drop $V_{23}$ at two points in the area between the current sourcing tips. By comparing 2-tip and 4-tip measurements, we determine the contact resistance to be ~1 kΩ, while the intrinsic sample resistance between the inner tips ~1 μm apart is $R_{12} \sim 40\,\Omega$. Compared with previous reports on nanofabricated devices[15], the observed resistance drop between the C and H states is much less pronounced because of the ~1 kΩ contact resistances of tips #1 and #2, and also because the current is not laterally confined on the surface, but flows over a wide area between the current-carrying tips (Fig. 2a). The MC calculations of the CLG model are described in Supplementary Note 6.

## Data availability

The data that support the findings of this study are available from the corresponding author upon request.

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

## Acknowledgements

We thank for the support from the Slovenian Research Agency (P1-0040, A.M. to PR-08972, A.K. to PR-06158, J.R. to PR-07589, J.V. to P1-0416 and J7-3146, V.K. to J1-2455, B.L. and M.T. to P2-0415) and Slovene Ministry of Science (Raziskovalci-2.1-IJS-952005). We wish to thank Serguei Brazovskii, Denis Golež and Vladimir Dobrosavljević for valuable discussions.

## Author contributions

A.M., Y.G., M.D., J.R., Y.V. and R.V. performed STM experiments; A.M., Y.G. and D.M. devised the experiments; A.K., J.V., V.K., P.K. and D.M. performed the theoretical analysis; B.L. and M.T. performed the thermal simulations; Y.V. provided some STM images; A.M., I.V. and D.M. wrote the paper. All authors revised and approved the paper.

## Competing interests

The authors declare no competing interests.
