## [Peer Review File · Nature Communications]

REVIEWER COMMENTS

Reviewer #1 (Remarks to the Author):

In this paper, using a multiple-tip scanning tunneling microscope, the authors investigated how the metastable network of dislocations in 1T-TaS₂ changes with increasing lateral electric current. The WS cell construction on the STM images showed the evolution of the network of dislocations during an 'erase' sequence, suggesting that the change in macroscopic electrical resistance arose from dislocation annihilation at high electric currents. Overall, this manuscript is well presented. I think the subject is interesting and of fundamental importance for manipulating metastable dislocation networks through carrier injection, with potential applications to memory and memristor. I support its publication after the following issues are appropriately addressed.

Comment 1.

While a rapid change in the slope dV/dI occurred at a threshold current of 2.0 mA, actual domain reconfiguration was observed when the current exceeded 3.2 mA. What is responsible for the mismatch in the two threshold currents? Contrary to the claims made in the manuscript, the dislocation annihilation does not seem to correlate directly with the change in electrical resistance.

Comment 2.

According to the discussion on the possible role of lattice imperfections ("Each time the experimental cycle is repeated, ..."), they repeated the "write" and "erase" sequences for a given sample. But, they presented results for only one cycle. I recommend the authors also show the data for repeated cycles in the SI to show the changes in the observed pattern during the cycles.

Comment 3.

To my understanding, the authors used the WS construction to "assign" fractional charges to dislocations in the sample rather than directly detecting the fractionalization through STM. As the authors mentioned, the fractionally charged quasiparticles cannot be individually removed, as this would violate topological rules. Then, what is the physical consequence of having fractional charges? Is there a way to detect them through transport measurement or other means? I wonder if the "fractional" charge provides a convenient way to classify dislocations or if it has a physical meaning on its own.

A recent theoretical study showed that twisted bilayer graphene with certain underlying symmetries hosts topological corner states with a fractional charge of $e/2$, which can also be observed in real space [M. J. Park et al., Phys. Rev. Lett. 123, 216803 (2019)]. In this case, the topological corner states can be detected through the quantum tunneling of electrons between them [M. J. Park et al., Carbon 174, 260 (2021)].

Comment 4.

In the structure analysis, the authors defined a filling fraction for site i as $f_i = f_c * A_i/A_c$. If I understand correctly, it should read $f_i = f_c * A_c/A_i$. Please check if there was a typo.

Reviewer #2 (Remarks to the Author):

The manuscript presents an experimental investigation of the current pulse-induced structural changes of the domain-like "hidden state" (H state) of the well-known charge-density-wave (CDW) system 1T-TaS₂. The changes in the network of dislocations and domain walls are induced and imaged along a write and erase cycle using a multi-tip scanning tunneling microscope. The observed defect dynamics and network topology changes are reproduced based on a charge-lattice-gas model, which has previously contributed to the qualitative understanding of the global phase diagram of CDWs in transition-metal dichalcogenides.

The results shown are interesting and relevant. Interesting because they provide a direct view of the current pulse-induced changes of a complex, spatially inhomogeneous CDW state that defies conventional theoretical description approaches such as Landau theory. Relevant because they suggest the possibility of selective control of this complex CDW state. The data shown are also of high quality, and the analysis and modeling of the data is robust, careful, and convincing.

My reservations about publishing the manuscript in its present form relate primarily to three aspects:

(1) Novelty

The study combines well-known experimental and theoretical approaches to study the well-known and already extensively studied H state in 1T-TaS₂: generation and manipulation by current pulses,

imaging by STM, and modeling by a charge-lattice-gas model. Bringing these approaches together is very compelling, but it remains unclear exactly where the experimental or theoretical innovation lies. I would therefore encourage the authors to elaborate more clearly on the truly novel aspects of the work.

(2) Mechanisms

The phenomenological agreement between the measured and simulated defect and network structures is impressive (Figures 3 and 4). However, the questions raised about microscopic mechanisms remain largely unanswered: What is the microscopic origin of the H state? How exactly does the formation and modification of the complex defect and domain structure proceed at the microscopic level? What is the origin of the metastability of this state? How exactly does charge transport work through this structure? Where exactly do the resistance jumps come from when the network configuration changes? Why is the observed phenomenology of network formation universal? It would be nice if the authors stated more clearly the new microscopic insights that their results provide with respect to finding answers to the open questions.

(3) Presentation

From my point of view, the text has a tendency to exaggerate or "oversell". The title, abstract, introduction, and conclusion play with terms like topological entanglement, topological protection, fractional charge, or Wigner crystal, which do not seem to fit 100% here, and in some cases may even be misleading. For example, what is the role of topology beyond topological defects? What is the origin of global topological protection? Does (quantum mechanical) entanglement really play a role here? Are only certain fractional charges allowed? Which ones? And can these in principle not be tuned continuously? Since all processes take place on a crystal lattice, it cannot be a "true" Wigner crystal, but a generalized Wigner crystal. Etc.

In summary, this is a manuscript that presents interesting and relevant results on a complex "hidden" CDW state in the context of a robust and convincing analysis. However, in my view, the methodological and scientific novelty of the results remains doubtful, the mechanistic explanations are rather generic, and the presentation in the text is too exaggerated. Therefore, I cannot recommend the manuscript in its current form for publication in Nature Communications.

Reviewer #3 (Remarks to the Author):

In this manuscript, Anze Mraz et al. report a combined study of multiple-tip scanning tunnel microscopy and model calculations on 1T-TaS₂. By applying the lateral electrical currents, the authors can induce the transitions between the commensurate insulating phase (C phase) and the hidden metastable phase (H phase). The hidden metastable phase contains the network of dislocations interconnected by domain walls. They can also induce the annihilation of dislocations by the electrical currents. Then they model the process as the “polaronic” Wigner crystal dynamics. Although the experimental data are interesting, I have several concerns about their measurements and their model. I cannot recommend the publication of this manuscript in Nature Communications. Below are my detailed comments.

- (1) By identifying the positions of the atomic defects, it seems to me that the STM image shown in Fig. 2c (0) is taken on a different area as the other STM images in Fig.2c. I suggest the authors replace the “0” state STM image with the one taken on the same area as in Fig. 2c (1-6).
- (2) For all the STM images shown in Figs. 1 and 2, the authors should write down the imaging conditions in the Figure captions, including the bias voltages and tunneling currents.
- (3) The authors show the evolution of the domain wall network with different lateral currents. After sending one current pulse, if the authors keep applying the same current pulse, will the network keep changing, or it stays at a fixed configuration with the fixed lateral current pulse setpoint?
- (4) The authors think that the evolution of the network in the H phase is due to the carrier injection. How can they exclude the possibility that this is not induced by the electric field between the STM tips 1 and 2? How does the duration of the current pulses influence the transition dynamics of the domain wall network?
- (5) The authors can perform the write and erase operations to the H phase by the lateral current. Does the sample completely recover back to the original “insulating” C phase? I suggest the authors check this by the local dI/dV measurements on the same area before and after the erase sequence. This could also show the influence of the lateral current induced buckling to the local electronic states.
- (6) In lines 79-85, the authors claim that STM can be used to investigate microscopically how $\rho(r)$ patterns evolve. At each point, the tunneling current is the integral of the local density of states from Fermi level to the bias voltage used for tunneling. It is not correct to take the STM image as the real-space charge density map.
- (7) In their model, they claim that the charged network comprises topologically trivial and non-trivial vertices connected by domain walls. To be clear, the authors should mention how they define topologically “trivial” and “non-trivial” vortices in the manuscript or the supplementary information?

RESPONSE TO REVIEWER COMMENTS

Reviewer #1 (Remarks to the Author):

In this paper, using a multiple-tip scanning tunneling microscope, the authors investigated how the metastable network of dislocations in 1T-TaS₂ changes with increasing lateral electric current. The WS cell construction on the STM images showed the evolution of the network of dislocations during an 'erase' sequence, suggesting that the change in macroscopic electrical resistance arose from dislocation annihilation at high electric currents. Overall, this manuscript is well presented. I think the subject is interesting and of fundamental importance for manipulating metastable dislocation networks through carrier injection, with potential applications to memory and memristor. I support its publication after the following issues are appropriately addressed.

Authors' response: We thank the referee for his/her opinion of the manuscript and the insightful remarks and suggestions, and respond to the issues below.

Comment 1.

While a rapid change in the slope dV/dI occurred at a threshold current of 2.0 mA, actual domain reconfiguration was observed when the current exceeded 3.2 mA. What is responsible for the mismatch in the two threshold currents? Contrary to the claims made in the manuscript, the dislocation annihilation does not seem to correlate directly with the change in electrical resistance.

Authors' response: The area which is imaged ($\sim 40 \times 40 \text{ nm}^2$) represents only a very tiny part of the switched area between the contacts that are $\sim 2.7 \mu\text{m}$ apart (Fig. 2a and SI-Fig. 3). If a semi-circular current path shape between the outer tips is assumed, we can estimate that we are observing $\sim 0.03\%$ of the entire switched area. The spatial variations of the network structure are quite significant on different length scales, presumably due to sample strains and imperfections, which means that the switching current threshold varies depending on the exact current paths. In fact, it would be unusual if the macroscopic measurement of current between two tips $2.7 \mu\text{m}$ apart showed precisely the same threshold as the microscopic reconfigurations on a small scale ($40 \times 40 \text{ nm}^2$). However, we note that we can observe a small change of slope in dV/dI in Fig. 2b also at current 3.2 mA, which is most likely linked to the observed microscopic changes recorded with the STM. We have slightly rephrased and added a few sentences in the MS to remove ambiguity, as well as a sentence in Section 2 of the Supplementary Information.

Comment 2.

According to the discussion on the possible role of lattice imperfections ("Each time the experimental cycle is repeated, ..."), they repeated the "write" and "erase" sequences for a given sample. But, they presented results for only one cycle. I recommend the authors also show the data for repeated cycles in the SI to show the changes in the observed pattern during the cycles.

Authors' response:

The original experiment included additional "write" and "erase" operations, which are now presented in the revised SI in Section 3 and SI-Fig. 4. The images reveal the difference of the domain wall patterns between different switching events, indicating that defects in the lattice most likely do not play a significant role in determining the pattern of the domain wall formation.

Besides that, we also repeated the experiment on a different 1T-TaS₂ sample and included the new images in the revised SI, where we can observe the difference between the

“written” state (SI-Fig. 6, H state), and the “re-written” state (SI-Fig. 6, 17.0 mA). We note that the measurements reveal a preferential direction for the domain walls along the direction of current, and an accompanying strain, which is visible as light streaks along the direction of the current in the topographic images (Fig. 2c and SI-Fig. 2 and SI-Fig. 4). Since this is an important effect, we make a note of this in the revised MS.

Comment 3.

To my understanding, the authors used the WS construction to “assign” fractional charges to dislocations in the sample rather than directly detecting the fractionalization through STM. As the authors mentioned, the fractionally charged quasiparticles cannot be individually removed, as this would violate topological rules. Then, what is the physical consequence of having fractional charges? Is there a way to detect them through transport measurement or other means? I wonder if the “fractional” charge provides a convenient way to classify dislocations or if it has a physical meaning on its own.

Authors' response: These are intriguing and very interesting questions. Indeed, we infer the texture by a WS construction, where the deduction of fractionalization comes from comparisons with the model calculations. Beyond a convenient classification of defects, the process for annihilation dynamics is revealed. Above all, it reveals the origin of the domain structure, and the nonlinear and highly non-trivial nature of the doping.

Regarding detection, as mentioned in our paper, a fractional charge cannot be removed (in a classical measurement only a single charge can be added or removed). A partial analogy exists with quarks, which carry fractional charge, but cannot be removed individually. The charge can be detected as ‘Barkhausen’ noise in response to excitation by electrical stimulus, or thermal excitation. However, we note that the topological defects themselves have a non-trivial quantum phase. A quantum calculation that models this would be of great interest. We note that the complexity of the domains and crossings in the $1/13$ filled superlattice structure may make such a calculation quite intricate, simply because there are many possible permutations. A deeper physical meaning of the fractionalization might be revealed in tunnelling experiments, but we would require model predictions in order to undertake them. We hope to address this in the future.

A recent theoretical study showed that twisted bilayer graphene with certain underlying symmetries hosts topological corner states with a fractional charge of $e/2$, which can also be observed in real space [M. J. Park et al., Phys. Rev. Lett. 123, 216803 (2019)]. In this case, the topological corner states can be detected through the quantum tunneling of electrons between them [M. J. Park et al., Carbon 174, 260 (2021)].

Authors' response: We thank the referee for pointing out very interesting publications showing tunnelling experiments between corner states in graphene. In comparison with graphene, the main challenge comes from the geometrical complexity of the superlattice texture in relation to the crystal structure. Furthermore, at present there is still a fundamental debate regarding the nature of interlayer interactions. Performing tunnelling experiments between corner states as reported by Park et al on the present material would require fabrication of appropriate structures with 1T-TaS₂. While this is feasible in principle, a theoretical prediction would be desirable before embarking on such a study. We hope that this may be possible in the future, but regrettably we do not have the possibility to include it at present. However, we make a note in the revised MS that might stimulate such an effort.

Comment 4.

In the structure analysis, the authors defined a filling fraction for site i as $f_i = f_c * A_i/A_c$. If I understand correctly, it should read $f_i = f_c * A_c/A_i$. Please check if there was a typo.

Authors' response: Indeed, this is a typo. Thank you for pointing it out.

Reviewer #2 (Remarks to the Author):

The manuscript presents an experimental investigation of the current pulse-induced structural changes of the domain-like "hidden state" (H state) of the well-known charge-density-wave (CDW) system 1T-TaS₂. The changes in the network of dislocations and domain walls are induced and imaged along a write and erase cycle using a multi-tip scanning tunneling microscope. The observed defect dynamics and network topology changes are reproduced based on a charge-lattice-gas model, which has previously contributed to the qualitative understanding of the global phase diagram of CDWs in transition-metal dichalcogenides.

The results shown are interesting and relevant. Interesting because they provide a direct view of the current pulse-induced changes of a complex, spatially inhomogeneous CDW state that defies conventional theoretical description approaches such as Landau theory. Relevant because they suggest the possibility of selective control of this complex CDW state. The data shown are also of high quality, and the analysis and modeling of the data is robust, careful, and convincing.

Authors' response: We thank the referee for his/her comments and for pointing out the relevance of the work and praising the data, analysis and modelling.

My reservations about publishing the manuscript in its present form relate primarily to three aspects:

(1) Novelty

The study combines well-known experimental and theoretical approaches to study the well-known and already extensively studied H state in 1T-TaS₂: generation and manipulation by current pulses, imaging by STM, and modeling by a charge-lattice-gas model. Bringing these approaches together is very compelling, but it remains unclear exactly where the experimental or theoretical innovation lies. I would therefore encourage the authors to elaborate more clearly on the truly novel aspects of the work.

Authors' response: We thank the referee for expressing his/her reservations and the encouragement to more clearly emphasize the truly novel aspects of the work. In response to the referee's suggestion, we have carefully modified the MS in the Introduction, Results and Discussion according to our response below:

In accordance with the editorial guidelines, we have not used statements emphasizing that something was done for the first time, but since there are many novelties, we list them here for the referee to consider:

- 1. Multi-tip experiments launching a surface current through two closely spaced tips and measuring microscopic charge rearrangements is reported for the first time in any**

material or device. It is experimentally extremely challenging and requires the building of a special apparatus, and to our knowledge has not been reported before. (STM imaging is itself well-known tool, but the current flows into the bulk and no information on CDW sliding, or the mechanism itself is directly obtained.)

2. Secondly, the experiment shows for the first time unambiguously that uniform CDW sliding due to applied field— a concept that has been valid for many decades- does *not* take place. Instead, it points to a new mechanism of domain wall (DW) and domain creation and topological physics caused by injection of charges. The experiment for the first time presents quantitative microscopic information on the topological dynamics in response to external charge injection.
3. This is the first time that STM heating effects are modelled, allowing the heating due to the surface current to be analysed, unambiguously showing that effect of the applied current on the domain creation and erasure is non-thermal, since the NC transition temperature is not reached even for the highest currents. Previous STM experiments attributed domain formation to a thermal effect (without proof).

Regarding the theoretical modelling:

1. Here for the first time the dynamics of the fractionalized charge state resulting from charge injection is revealed, starting with a single electron, extending eventually to a network in which the charges are distributed over the domain walls and are intricately entangled with each other. The model shows impressive agreement with experiment.
2. The resulting concept of fractional charge is thus introduced. The concept is known theoretically from nuclear quantum chromodynamics (QCD) and one-dimensional systems such as polyacetylene chains. While in 1D, the dynamics of discommensurations are described in terms of transitions on 'Devil's staircase' solutions which represent the Cantor set solution of the Frenkel-Kontorova model¹, the model is not easily tractable in 2D²⁻⁴. Here we present the CLG model, with remarkable success, which shows explicitly how injecting charge, one by one, leads to the creation of fractionally charged networks.

(2) Mechanisms

The phenomenological agreement between the measured and simulated defect and network structures is impressive (Figures 3 and 4).

Authors' response: We agree that the model simulations of the effect of injected charge impressively reproduce the measured network structure, and we consider the charge-injection mechanism that creates the observed topologically non-trivial mesoscopic structures to be a convincing mechanism for producing the observed topologically protected structure.

However, the questions raised about microscopic mechanisms remain largely unanswered: What is the microscopic origin of the H state?

Authors' response: The microscopic mechanism is subject to symmetry and topological constraints. We believe that it is clear that the formation of a topologically protected spatial network defines the microscopic structure, and not vice versa. (see below)

We thus respectfully disagree that the mechanism remains largely unanswered. We believe, in agreement with the other two referees, that the mechanism is topological in nature.

How exactly does the formation and modification of the complex defect and domain structure proceed at the microscopic level?

Authors' response: Calculating the total energy of each domain configuration is conceptually trivial, but practically intractable as far as we are aware. To calculate reliable microscopic structures that show a false vacuum state self-consistently would require some kind of structural relaxation with thousands of atoms within the appropriate unit cell encompassing

entire domains and domain walls, for different domain structures. Then we would need to compare the stability of different domain structures and eventually different stackings to find which configuration is energetically favourable amongst myriads of possibilities. We suggest that this is not tractable at present.

What is the origin of the metastability of this state?

Authors' response: Our view is that the origin of the metastability is revealed to be the creation of topologically protected states. Fundamentally this is the same as the vortices and other topological solitons that give rise to quasi-stable “elementary particles” in dense quark matter. A nice overview of topological objects in QCD that are analogous to the present examples is given by Eto et al.⁵

How exactly does charge transport work through this structure?

Authors' response: This question was already discussed in the original manuscript, albeit briefly. The DWs and vertices were previously shown by us and other authors to be gapless quite early on, but have different local structure. The domains themselves may be metallic or insulating, as suggested by DFT calculations. We have argued that DFT calculations appear to show that c-axis stacking of the charge order has important bearing on the transport properties, keeping in mind that the c-axis stacking of a domain state is imposed through topological rules that define the network structure. Recognizing the importance of this issue, we have included some more discussion (in the Discussion section) and a new panel in Figure 4 (p-t) as a help to understanding.

Where exactly do the resistance jumps come from when the network configuration changes?

Authors' response: This issue was already discussed early on, in the context of macroscopic measurements of thermal resistivity relaxation of the H state (ref. 12 in the MS). The resistance depends on the domain configuration, irrespective of whether inter-layer transport is involved or not. With the new data available here, in the discussion we have addressed the problem of transport which depends on the c-axis stacking (see below) in terms of the suggestions from DFT calculations that imply different band structures for differently stacked bilayers, with a schematic figure (mentioned above). However, the subject of out-of-plane resistance in relation to interlayer stacking is still controversial, and we have some reservations regarding this issue. Solving this controversy is not the subject of the present paper, but we argue that FIB-fabricated samples in recent experiments are most likely strained⁶, which leads to conducting channels around the edges. (Multiple experiments have shown that strain strongly changes the transport properties of the material^{1,7}). Thus, it is not yet clear what is the mechanism of c axis transport in the C state. The presence of the mosaic of the H state, where domains in different layers overlap only sporadically further complicates the issue. This is mentioned in the Discussion of the revised MS.

Why is the observed phenomenology of network formation universal?

Authors' response: The texture formation in a polaronic Wigner crystal upon doping applies universally to any sparsely-filled system of localized charges on a triangular lattice that satisfies the geometric cosine rule (as stated). Similar textures have been theoretically predicted and also experimentally observed for *doped* sparsely filled lattices with filling fractions near $1/4$, $1/7$, $1/9$, $1/13$ and $1/16$. In this sense the phenomenology of network formation is universal, which has been experimentally confirmed (ref. 28 in the MS). On the other hand, switching between insulating and metallic charge transport was not yet shown to be universal, and DFT calculations suggest that this depends on the detailed band structure of differently stacked CDW multilayers.

It would be nice if the authors stated more clearly the new microscopic insights that their results provide with respect to finding answers to the open questions.

Authors' response: We thank the referee for this suggestion. In the revised MS, we address more clearly the microscopic level insights in the context of the topological constraints that

are derived from the present work, particularly related to transport properties. We have already presented some discussion of how charge transport takes place, based on current understanding, as discussed above.

(3) Presentation

From my point of view, the text has a tendency to exaggerate or "oversell". The title, abstract, introduction, and conclusion play with terms like topological entanglement, topological protection, fractional charge, or Wigner crystal, which do not seem to fit 100% here, and in some cases may even be misleading.

Authors' response: We carefully considered the terminology, using the most appropriate language to describe the relevant phenomena to avoid confusion. There was absolutely no intention to exaggerate or 'oversell'. These concepts are crucial aspects of the present work, and we believe it is better to use established and unambiguous language than invent new terminology, unless absolutely necessary.

We justify and explain the issues raised:

Topological entanglement:

In the text we emphasized that the entanglement that we discuss is classical. The charge is entangled in the sense that a single charge cannot be removed without affecting the network on an extended scale. Conversely, the addition of a single charge results in an extended highly non-trivial modification of the network. This makes the system highly non-linear with respect to doping. There is no need to shy away from using the term entanglement, its use is justified and appropriate.

For example, what is the role of topology beyond topological defects?

Authors' response: Topology defines the rules for the formation of the observed extended networks (domain walls and vertices). The entire structure is topologically protected, and the topological defect have non-local effects.

What is the origin of global topological protection?

Authors' response: The origin of global topological protection comes from the fact that the network cannot be changed locally, without a global, topologically non-trivial transformation that acts on the entire network. This is a consequence of the fact that all charges are entangled with each other in the network. When an object is stretched or squashed — or otherwise deformed without being broken — and its features stay the same, the object is said to be "topologically protected." The topological rules are explicitly shown to be obeyed experimentally in the dynamics at long range. This is true globally for the networks discussed here.

Does (quantum mechanical) entanglement really play a role here?

Authors' response: Good question. For now, we do not discuss the quantum properties of the networks, which is beyond the scope of the present paper (see response to referee #1, point 3.) However, a moment's thought leads to the realization that the quantum properties of the system are highly non-trivial because of the topological structure of the mesoscopic networks. In the manuscript it was emphasized that we were discussing classical entanglement. Quantum entanglement and Berry phase of the constructions will be addressed by future work.

Are only certain fractional charges allowed? Which ones? And can these in principle not be tuned continuously?

Authors' response: This is a very interesting question; we are grateful that it was raised, which gives a chance to discuss it. For a finite number of injected charges N , the number of fractional Wigner-Seitz (WS) polygons (in which the charge per WS cell departs from $1/13$) is finite, and so is the number of possible surface areas (i.e. charge densities). This is a

consequence of the geometric constraints on the WS constructions imposed by the atomic lattice. Hence, even though there is a large number of possible fractional charge densities (WS cell areas), fractional charge cannot be tuned continuously, but must follow the geometric rules. We illustrate this question with an example, by including a figure in the Supplemental Information (Section 7, SI-Fig. 9) which shows the fractional charge of each WS cell for the case of a single injected electron ($N=1$). A comment is also added to the MS.

Since all processes take place on a crystal lattice, it cannot be a "true" Wigner crystal, but a generalized Wigner crystal. Etc.

Authors' response: Here we discuss *polaronic* Wigner crystals (WCs): There is a difference between the original WC as discussed theoretically by Wigner and the polaronic WC. As discussed in the introduction of this paper (and other publications), polaronic effect leads to a band narrowing and a large ratio $r_s = V/t$, which in turn leads to carrier localization due to Coulomb repulsion as Wigner envisaged it. We note that the present modelling is classical. Quantum models are still being developed⁸⁻¹⁰.

In summary, this is a manuscript that presents interesting and relevant results on a complex "hidden" CDW state in the context of a robust and convincing analysis. However, in my view, the methodological and scientific novelty of the results remains doubtful, the mechanistic explanations are rather generic, and the presentation in the text is too exaggerated. Therefore, I cannot recommend the manuscript in its current form for publication in Nature Communications.

Authors' response: We respectfully disagree regarding the scientific novelty of the results. We note that it is precisely the purpose of mechanistic descriptions to reveal the salient features of a problem that address a general problem without dwelling on unnecessary details. I believe that we have succeeded in this respect. The referee's own words state that the agreements between experiments and the model are impressive.

The fact that the presented novel 'mechanistic explanations' are generic means that they can be applied to a wide variety of systems, and there is some indication in the literature that this is indeed the case. However, the mechanistic explanation is really a topological one, which goes much deeper than microscopic theory can provide because it gives constraints which microscopic theory must eventually deal with.

In response, we have carefully addressed the presentation, highlighted the conceptual novelty more explicitly, and carefully addressed the various issues that were raised.

Reviewer #3 (Remarks to the Author):

In this manuscript, Anze Mraz et al. report a combined study of multiple-tip scanning tunnel microscopy and model calculations on 1T-TaS₂. By applying the lateral electrical currents, the authors can induce the transitions between the commensurate insulating phase (C phase) and the hidden metastable phase (H phase). The hidden metastable phase contains the network of dislocations interconnected by domain walls. They can also induce the annihilation of dislocations by the electrical currents. Then they model the process as the "polaronic" Wigner crystal dynamics. Although the experimental data are interesting, I have several concerns about their measurements and their model. I cannot recommend the publication of this manuscript in Nature Communications. Below are my detailed comments.

Authors' response: we thank the referee for his/her remarks and address the concerns below:

(1) By identifying the positions of the atomic defects, it seems to me that the STM image shown in Fig. 2c (0) is taken on a different area as the other STM images in Fig.2c. I suggest the authors replace the “0” state STM image with the one taken on the same area as in Fig. 2c (1-6).

Authors' response: After the sample was switched, the scanning area was slightly moved (by ~50 nm) to obtain some identifiable defects in the polaronic lattice to track the registry. We thus do not have the image of exactly the same area in the C state to replace Fig. 2c, panel 0. In response to the referee, we now repeated the experiment on a different 1T-TaS₂ sample. The images are presented in the revised SI (SI-Fig. 4 and SI-Fig. 5), where W-pulse switching between C and H state was recorded on the same spot. The experiments fully confirm the statements made in the original MS.

(2) For all the STM images shown in Figs. 1 and 2, the authors should write down the imaging conditions in the Figure captions, including the bias voltages and tunneling currents.

Authors' response: We added a general note about this in the Methods section in the MS and in the Figure captions where applicable.

(3) The authors show the evolution of the domain wall network with different lateral currents. After sending one current pulse, if the authors keep applying the same current pulse, will the network keep changing, or it stays at a fixed configuration with the fixed lateral current pulse setpoint?

Authors' response: Even when applying current pulses with amplitude well below the switching threshold ($\ll 3$ mA), the domain wall configuration changes ever so slightly after each pulse (even only 1 polaron movement). This is also true when we apply the same current pulse multiple times, although this was not systematically investigated.

(4) The authors think that the evolution of the network in the H phase is due to the carrier injection. How can they exclude the possibility that this is not induced by the electric field between the STM tips 1 and 2? How does the duration of the current pulses influence the transition dynamics of the domain wall network?

Authors' response: One clear indication that charge injection is responsible for the formation of the H phase is that it also forms in response to photo-injection of charge, where no static field is present. This was discussed in ref. 11 in the MS. An electric field is required to set up a current. In this sense, the two cannot be separated. In our experiments, we use a constant current source. However, the sample has a finite resistance which is changing with time, so it is difficult to completely separate the effect of the electric field from the current. However, the expected effect of the field alone would be to cause sliding of the C-CDW, not the creation of a H state. Our experiments show that sliding is conspicuously absent, as noted in the MS.

The effect of the pulse length on the resistance switching was presented recently in another study (ref. 15 in the MS). We did not systematically study the effect of pulse length in the present imaging experiments.

(5) The authors can perform the write and erase operations to the H phase by the lateral current. Does the sample completely recover back to the original “insulating” C phase? I suggest the authors check this by the local dI/dV measurements on the same area before and after the erase sequence. This could also show the influence of the lateral current induced buckling to the local electronic states.

Authors' response: This is an interesting suggestion. We would like to point out the experimental complexity of the three-tip STM setup and of the performed measurement, where unfortunately obtaining the proposed dI/dV measurements was not possible because the tip became unstable after the passing of current during the duration of the experiment. However, we can still say that in our original experiment (shown in Fig. 2 and SI-Fig. 2) the sample did not recover fully to the pristine bulk C state within our small scan area and a domain wall remains visible. This is also true for our repeated experiment (SI-Fig. 6), where additionally it seems that we can also discern a charge density wave mosaic on the bottom layer. The observations are fully consistent with the discussion and observations presented originally in the main paper.

We would like to further note that in the context of practical memory devices on 1T-TaS₂ with fabricated electrodes^{11,12}, complete recovery of the insulating state is also probably not achieved, since the maximum resistance value of the device is usually reached only after the initial cooling down from room temperature, and then rarely reached again with electrical switching at low temperatures (<20 K). However, the performance of such devices is still very remarkable with over 10⁶ W/E cycles between the high and low resistance state with very good stability of the two states. This probably means that certain mosaic patterns are formed somewhere in the structure, which over time are no longer influenced by additional pulsing (resembling “training” in memristors), most likely associated with some pinning of the domain wall structure in the vicinity of the fabricated contacts, or jammed tips in our case. To completely relax the charge configuration structure back to the pristine C state it seems that full thermal cycling is required.

(6) In lines 79-85, the authors claim that STM can be used to investigate microscopically how $\rho(r)$ patterns evolve. At each point, the tunneling current is the integral of the local density of states from Fermi level to the bias voltage used for tunneling. It is not correct to take the STM image as the real-space charge density map.

Authors' response: Here, some clarification is needed. Indeed, it is conventionally derived, under certain assumptions, that the tunnelling current is proportional to the integral of the local density of states from the Fermi level to the bias voltage (and the matrix element). For a polaronic Wigner crystal (or CDW), the tunnelling current is position dependent and reflects not only the spatial variation of the LDOS, but also the charge density. The reason for this is that when polaron forms, the band structure is locally deformed to accommodate the single electron at the centre of the polaron, which changes the LDOS. Hence there is a direct link between the presence of the polaron's LDOS and the presence of an electron at the centre of the polaronic cluster (Fig. 1). For the purposes of analysing the present experiments, this connection is sufficient, because we use it to detect *the presence or absence of polarons*. We note that the structure of polarons in the DWs and crossings are modified, but we do not consider these in detail in the present paper. No doubt it is worth investigating this in detail in future, as was done by Yeom's group for example¹³. In the main text of the revised MS we clarify this point, and include a more detailed explanation in the Supplemental Information, Section 5.

(7) In their model, they claim that the charged network comprises topologically trivial and non-trivial vertices connected by domain walls. To be clear, the authors should mention how they define topologically “trivial” and “non-trivial” vortices in the manuscript or the supplementary information?

Authors' response: We apologize for not properly defining the terms. In the revised version of the article, we define a trivial vortex to be one for which no dislocation is formed in the Wigner-Seitz superlattice, and the Burger's vector is zero. A non-trivial vortex is one which forms a dislocation on the WS superlattice, with a non-zero Burger's vector. We note that this structure appears in the model as well as the WS tessellation of the STM data.

1. Vaskivskiy, I. *et al.* Controlling the metal-to-insulator relaxation of the metastable hidden quantum state in 1T-TaS 2. *Sci. Adv.* **1**, e1500168 (2015).
2. Bak, P., Mukamel, D., Villain, J. & Wentowska, K. Commensurate-incommensurate transitions in rare-gas monolayers adsorbed on graphite and in layered charge-density-wave systems. *Phys Rev B* **19**, 1610–1613 (1979).
3. Ravnik, J., Vaskivskiy, I. & Gerasimenko, Y. Strain-Induced Metastable Topological Networks in Laser-Fabricated TaS₂ Polytype Heterostructures for Nanoscale Devices. *Acs Appl Nano Mater* **2**, 3743–3751 (2019).
4. Villain, J. & Bak, P. Two-dimensional ising model with competing interactions : floating phase, walls and dislocations. *J Phys-paris* **42**, 657–668 (1981).
5. Eto, M., Hirono, Y., Nitta, M. & Yasui, S. Vortices and other topological solitons in dense quark matter. *Prog. Theor. Exp. Phys.* **2014**, 012D01 (2014).
6. Martino, E. *et al.* Preferential out-of-plane conduction and quasi-one-dimensional electronic states in layered 1T-TaS₂. *Npj 2d Mater Appl* **4**, 7 (2020).
7. Svetin, D. *et al.* Transitions between photoinduced macroscopic quantum states in 1T-TaS₂ controlled by substrate strain. *Appl. Phys. Express* **7**, 103201 (2014).
8. Ravnik, J. *et al.* Quantum billiards with correlated electrons confined in triangular transition metal dichalcogenide monolayer nanostructures. *Nat Commun* **12**, 3793 (2021).
9. Fratini, S., Driscoll, K., Ciuchi, S. & Ralko, A. A quantum theory of the nearly frozen charge glass. *Scipost Phys* **14**, 124 (2023).
10. Vodeb, J. *et al.* Emergent false vacuum decay processes in a two-dimensional electronic crystal: experiment vs. simulations on a noisy superconducting quantum processor. (2021). at <<https://arxiv.org/abs/2103.07343>>;
11. Mraz, A. *et al.* Charge Configuration Memory Devices: Energy Efficiency and Switching Speed. *Nano Lett* **22**, 4814–4821 (2022).
12. Mihailovic, D. *et al.* Ultrafast non-thermal and thermal switching in charge configuration memory devices based on 1T-TaS. *APL* **119**, 013106 (2021).
13. Park, J. W., Lee, J. & Yeom, H. W. Zoology of domain walls in quasi-2D correlated charge density wave of 1T-TaS₂. *Npj Quantum Mater* **6**, 32 (2021).

REVIEWERS' COMMENTS

Reviewer #1 (Remarks to the Author):

In the revised manuscript, the authors have addressed almost all of my concerns and suggestions. However, I have two recommendations related to the presentation of the concepts of “fractional charges” and “classical entanglement.”

(1) I think the analogy with quarks (lines 272-275 on page 9) is misplaced and unnecessary. The confinement of fractionally charged quarks arises from the asymptotic freedom of a specific class of gauge theories and is not directly relevant to the indivisibility of fractionally charges in doped TaS₂. Therefore, I recommend removing the mention of “quarks” from the manuscript. Additionally, the physics of topological solitons discussed in dense quark matter (ref. 5 in the rebuttal letter) is unrelated to the topic addressed in the current paper.

(2) While the authors have introduced a sentence stating, “The charges are classically entangled ...” (lines 183-185 on page 6), I suggest emphasizing the “classical” nature of the topological entanglement and the model calculations at an earlier stage in the manuscript. Consider elaborating on this point to enhance the reader’s understanding and avoid any confusion.

Reviewer #2 (Remarks to the Author):

In their reply letter, the authors address all of the reviewers' criticisms in a detailed, thorough, and constructive manner. In particular, I thank them for their detailed answers to my relatively many questions. This made the scientific novelty, the microscopic mechanisms and the terminology much clearer to me. Overall, I find the answers convincing for the most part. Further discussion of points where there may still be disagreement would go too far.

The authors have also responded constructively to the reviewers' criticisms by revising the manuscript and providing supplementary information. The changes and additions are substantial, strengthen the findings, and facilitate a better understanding of the results and conclusions.

In my opinion, the manuscript has been significantly improved and I can now recommend it for acceptance.

Reviewer #3 (Remarks to the Author):

I think the authors have addressed the comments in my previous review report, and I have no further comments.

In the revised manuscript, the authors have addressed almost all of my concerns and suggestions. However, I have two recommendations related to the presentation of the concepts of “fractional charges” and “classical entanglement.”

Authors response: We thank the referee for his/her comments and recommendations. We accept the suggestions, and have modified the manuscript accordingly.

(1) I think the analogy with quarks (lines 272-275 on page 9) is misplaced and unnecessary. The confinement of fractionally charged quarks arises from the asymptotic freedom of a specific class of gauge theories and is not directly relevant to the indivisibility of fractionally charges in doped TaS₂. Therefore, I recommend removing the mention of “quarks” from the manuscript. Additionally, the physics of topological solitons discussed in dense quark matter (ref. 5 in the rebuttal letter) is unrelated to the topic addressed in the current paper.

Authors response: We could probably debate to what extent the systems are similar, and how far the analogies are useful, but we are perfectly happy to leave this discussion to another forum. Thus, we have followed the recommendation and removed the sentence: “Similarly as for quarks, fractionally charged quasiparticles cannot be individually removed, as this would amount to isolating a fractional charge, which is not possible without violating topological rules.”, as this indeed implies some similarity in the mechanism, which may indeed be misleading.

(2) While the authors have introduced a sentence stating, “The charges are classically entangled ...” (lines 183-185 on page 6), I suggest emphasizing the “classical” nature of the topological entanglement and the model calculations at an earlier stage in the manuscript. Consider elaborating on this point to enhance the reader’s understanding and avoid any confusion.

Authors response: This makes sense. We have added a brief introduction to classical entanglement in the last part of the introduction, where we describe the formation of topologically networks. We hope that this will enhance the reader’s understanding and avoid any possible confusion.